Assessing subspecies status of leopards (Panthera pardus) of northern Pakistan using mitochondrial DNA

Asad Muhammad muhammad.asad@lincolnuni.ac.nz 1
Martoni Francesco 2
Ross James G. 1
Waseem Muhammad 3
Abbas Fakhar-i- 4
Paterson Adrian M. 1
1 Department of Pest-management and Conservation, Faculty of Agriculture and Life Science, Lincoln University , Lincoln , Canterbury , New Zealand
2 AgriBio Centre for AgriBioscience, Agriculture Victoria Research , Bundoora , Victoria , Australia
3 World Wide Fund for Nature (WWF) , Islamabad , Pakistan
4 Bioresource Research Centre , Islamabad , Pakistan
Ray David
Electronic publication date: 2019 Jul 16
Publication date: 2019
Volume: 7
Electronic Location ID: e7243
Received 2019 Mar 12; Accepted 2019 Jun 3
Copyright: ©2019 Asad et al.
Copyright year: 2019
Copyright holder: Asad et al.
License: This is an open access article distributed under the terms of the Creative Commons Attribution License, which permits unrestricted use, distribution, reproduction and adaptation in any medium and for any purpose provided that it is properly attributed. For attribution, the original author(s), title, publication source (PeerJ) and either DOI or URL of the article must be cited.
License URL: https://creativecommons.org/licenses/by/4.0/

Keywords: Genetic variation, New haplotype, Distribution, P. p. saxicolor, P. p. fusca, Asian leopards

Funding: Lincoln University Research This work was supported by a Lincoln University Research grant. The funders had no role in study design, data collection and analysis, decision to publish, or preparation of the manuscript.

==============================
Despite being classified as critically endangered, little work has been done on leopard protection in Pakistan. Once widely present throughout this region, leopards are now sparsely distributed, and possibly extinct from much of their previously recorded habitat. While leopards show morphological and genetic variation across their species range worldwide, resulting in the classification of nine different subspecies, the leopard genetic structure across Pakistan is unknown, with previous studies including only a very limited sampling. To clarify the genetic status of leopards in Pakistan we investigated the sequence variation in the subunit 5 of the mitochondrial gene NADH from 43 tissue samples and compared it with 238 sequences available from online databases. Phylogenetic analysis clearly separates the Pakistani leopards from the African and Arabian clades, confirming that leopards from Pakistan are members of the Asian clade. Furthermore, we identified two separate subspecies haplotypes within our dataset: P. p. fusca (N = 23) and P. p. saxicolor (N = 12).

Introduction

Leopards (Panthera pardus) are classified as critically endangered in many habitats where they are present (Uphyrkina et al., 2001), including Pakistan (Sheikh & Molur, 2004). There are currently nine recognized subspecies of the common leopard (Uphyrkina et al., 2001) and two of these, African (P. p. pardus) and Indian (P. p. fusca), are considered “near threatened”, while the rest are classified as “critically endangered”, “endangered” and “near threatened to endangered” (Sheikh & Molur, 2004; Stein et al., 2016).

Globally, leopard subspecies recognition is based on genetic, morphological and geographical information (Jacobson et al., 2016). However, leopards show high genetic and morphological variation across their range and, in many cases, genetic patterns do not align with the geographical variation recorded for previously defined subspecies (Uphyrkina et al., 2001). Therefore, the study of the genetic structure of leopard populations is considered vital for a better understanding of both subspecies and population subdivision and, consequently, for their conservation (Khorozyan, Baryshnikov & Abramov, 2006; Sugimoto et al., 2013). Molecular studies have contributed significantly in the field of conservation for many other elusive cat species, such as snow leopards, lions and tigers (Dubach et al., 2005; Wei, Wu & Jiang, 2008; Bhavanishankar et al., 2013). Similarly, in the case of leopards, genetic analyses increasingly provide taxonomic guidance for subspecies identification (Uphyrkina et al., 2001; Arif et al., 2011; Sugimoto et al., 2014).

Conservation and management of leopards are difficult tasks made even harder by a general lack of understanding of the broad geographic ranges and adaptability of this species, factors that make leopards more detectable than their actual numbers would warrant (Jacobson et al., 2016). These detections increase the misconception that leopards are not as severely endangered as they actually are (Jacobson et al., 2016). In Pakistan, while leopards are known to be present, there is no information on their distribution, number or subspecies. Based on the severe decline rate of leopard populations in Asia, the Pakistani population is likely to be fragmented and with depleted genetic variation (Dutta et al., 2013; Jacobson et al., 2016). Asian leopards have lost around 83–87% of their former range, compared with a 48–67% decline in Africa (Sheikh & Molur, 2004; Khorozyan, Baryshnikov & Abramov, 2006; Laguardia et al., 2015; Jacobson et al., 2016). It is, therefore, essential to identify the subspecies present in Asia to prevent further loss of biodiversity (Arif et al., 2011).

Pakistan provides a major disjunction in the fauna and flora of southern Asia, divided along the western edge of the Indus Basin and the upper Indus valley of Kohistan (Frodin, 1984). In the past, this biogeographic disjunction has served as a geographical limit to two of the main standard floras for Southwest and South Asia: Boissier’s Flora Orientalis (1867–1888) and Hooker’s Flora of British India (1872–1897).

Four leopard subspecies have been suggested as present in Pakistan: Panthera pardus fusca, P. p. saxicolor, P. p. sindica, and P. p. millardi. These subspecies were identified based solely on morphological characters, such as unique coat pattern, coloration, fur length, body size and skull size (Pocock, 1930a; Pocock, 1930b; Roberts, 1977). However, these characters vary in different environmental conditions and may lead to subspecies misidentification (Uphyrkina et al., 2001). A morphological analysis of Pakistani leopard skulls (Khorozyan, Baryshnikov & Abramov, 2006) suggested the presence of only two subspecies: P. p. sindica from Baluchistan, similar to the subspecies population in southern Iran, and P. p. millardi from Kashmir, similar to the population present in India. A recent study based on the global geographic distribution patterns suggested that the subspecies present in the region were P. p. saxicolor and P. p. fusca (Jacobson et al., 2016). The only genetic sample collected from a Pakistani leopard in the wild was from a single individual in the region of Baluchistan, and identified as the subspecies P. p saxicolor (Uphyrkina et al., 2001). This study aims to better understand the genetic structure of the leopard in Pakistan, identify the subspecies present, and establish baseline information for future monitoring. We present results based on more than 40 leopard samples from Pakistan and we investigate genetic variation in the mtDNA, targeting the subunit 5 of the NADH gene (NADH 5), following the approaches of previous studies (e.g., Uphyrkina et al., 2001; Farhadinia et al., 2015; Anco et al., 2018). The choice of mtDNA was based on the good performances and wide adoption for felids (e.g., Jae-Heup et al., 2001; Luo et al., 2004; Barnett et al., 2006; Havird & Sloan, 2016). In particular, the choice of NADH 5 gene was based on the good performance of this gene in subspecies delimitation for carnivores, showing a higher mutation rate for this group (e.g., Lopez et al., 1997). Furthermore, considering the general use of NADH 5 in leopard studies (e.g., Uphyrkina et al., 2001; Farhadinia et al., 2015; Anco et al., 2018), we focused on this gene also for continuity with previous works, by generating sequences that can be added to previous datasets. More specifically, our samples are mostly sourced from the northern region of Pakistan (Galyat, Murree, and Aazad Kashmir) with a single specimen from Baluchistan. This research will add to the scarce existing scientific knowledge of species assessment in Pakistan, helping prioritize the conservation efforts for leopards, and will consequently contribute to the international works on felids.

Materials and Methods

Samples collection

Animal ethics committee approval AEC2017-02 was obtained prior to the experiment. A total of 49 samples of leopard skin tissue was initially obtained for this study. Most of these samples were from the northern region of Pakistan, where the leopard population is considered stable (Shehzad et al., 2014) (Fig. 1). Of these, 22 leopards were killed by communities in retaliation for attacks, including seven killed during the period of this study. Two leopards had died of natural causes, the rest of the leopard mortality is unknown. Eighteen samples were provided by the Bioresource Research Centre of Islamabad (Pakistan) and had been used in a previous study (Bebi et al., 2015). Nine tissue samples were provided by Wildlife Department Khyber Pakhtunkhwa and 15 samples from the World Wide Fund for Nature Pakistan (Fig. 1). The majority of the samples were collected from two different areas: the Azad Jamu Kashmir and the Galyat region. Additionally, two samples were provided by BRC from Baluchistan and Sukkur.

Figure 1 Sampling locations of Panthera pardus in the northern regions of Pakistan (Galyat and Azad Kashmir).

The 35 samples included in the haplotype analysis are represented here. Red dots indicate samples belonging to haplotype A; blue squares are samples from haplotype B; and the single Yellow triangle indicates haplotype C. The different colours present in the map indicate the habitat where samples were found, highlighting the presence of natural reserves, national parks and forests.

DNA extractions and amplifications

All tissues were preserved in 70% ethanol until DNA extraction and amplification.

In a preliminary analysis conducted at the Bio Resource Research Centre facilities of Islamabad (Pakistan), DNA was extracted from three samples using the QIA amp DNA Mini Kit (QIAGEN) following the manufacturer’s instructions while the remaining samples were sent to Macrogen Inc. (Seoul, Korea) for DNA extraction and polymerase chain reaction (PCR) optimization. Four samples were discarded because of a lack of detailed information on their origin, while two samples were not sequenced due to possible contamination during the transport.

Using the full mitochondrial sequence of leopard available on the National Center for Biotechnology Information (NCBI) database (accession number EF551002.1); two sets of primer pairs (F/RL2 and FL2/RL4; Table 1) were designed to target two overlapping regions of the subunit 5 of the NADH mitochondrial gene. These corresponded to nucleotide positions 12632-13242 of the mitochondrial DNA of leopards. Primer design, PCR amplification, PCR purifications and sequencing were outsourced to Macrogen Inc. (Seoul, Korea) and were carried out following the methods of previous studies (Uphyrkina et al., 2001; Farhadinia et al., 2015). Samples preparation and DNA extraction were conducted in a laminar flow hood in an area isolated from other samples to prevent any contamination. Genomic DNA was isolated using InstaGene Matrix (Bio-Rad Laboratories, Hercules, CA, USA) and MG Tissue SV (Doctor protein inc, Korea). PCR was run for 35 cycles, with 5 min pre-denaturation at 95 °C followed by denaturation at 94 °C for 30 s, and 30 s annealing at 50 °C, and by 1 min extension at 72 °C. Products were then checked in 1.5% agarose gel, running for 20 min at 300V, 200A. Purification was carried out with multiscreen filter plate (Millipore Sigma, Burlington, MA, USA). Each PCR product obtained was sequenced (both forward and reverse) by Macrogen Inc. (Seoul, Korea) using Sanger sequencing technologies.

Table 1 Primers used for DNA isolation and amplification.

Name, direction and sequences of the primers designed for this study are reported.

Primer Name	Direction	Sequence (5′ to 3′)	
F	Forward	GTGCAACTCCAAATAAAAG	
RL2	Reverse	TAAACAGTTGGAACAGGTT	
FL2	Forward	CGTTACATGATCGATCATAG	
RL4	Reverse	TTAGGTTTTCGTGTTGGGT	

Data analysis

A total of 43 samples produced viable sequences. Sequencing data is available at NCBI GenBank via accession numbers MK425702–MK425744. This data is also available as a Supplemental File. The software MEGA 7 (Kumar, Stecher & Tamura, 2016) was used to visually inspect the electropherograms, to align the sequences and generate a pairwise distance matrix that was then used to identify the different haplotypes.

In order to compare the haplotypes obtained here with other sequences belonging to the species Panthera pardus, all 238 sequences present on the GenBank database (to the month of August 2018) were obtained (Table 2) and aligned with those collected in this study together with sequences from snow leopards, tigers and lions as outgroups (Table 2). We used the software MEGA X (Kumar et al., 2018) to identify the best model of nucleotide substitution based on the Bayesian Information Criterion (BIC). This reported the Kishino and Yano (HKY) + G model of nucleotide substitution (gamma distribution with five rate categories) as the best model, as previously found elsewhere (Farhadinia et al., 2015). Additionally, the Kimura-2-parameters (K2P) substitution model was also tested here. A maximum likelihood (ML) NADH 5 gene tree was generated using MEGA X, setting the bootstrap to 10,000 replicates and using both the substitution models. PopArt (Leigh & Bryant, 2015) was used to construct a Median Joining haplotype network (ε = 0) of all the sequences from Asian leopards.

Table 2 Leopard sequences obtained from GenBank together with outgroups used in our study.

Accession numbers are reported with the number of sequences for each gene region per species and subspecies used in this work. Origin of the samples and original work publishing these sequences are reported.

Accession numbers	Sequences	Gene	Species	Subspecies	Sample area	Reference	
KY292222.1–KY292277.1	56	NADH 5	Panthera pardus	Africans	Africa	Anco et al. (2018)	
JX559073.1–JX559076.1	4	NADH 5	P. pardus	Saxicolor	Iran	Farhadinia et al. (2012, unpublished)	
JF720187.1–JF720319.1	133	NADH 5	P. pardus	LEO(African)	Africa	Ropiquet et al. (2015)	
HQ185549.1–HQ185550.1	2	NADH 5	P. pardus	orientalis	Caucasus	Rozhnov, Lukarevsky & Sorokin (2011)	
HQ185544.1–HQ185548.1	5	NADH 5	P. pardus	saxicolor	Caucasus	Rozhnov, Lukarevsky & Sorokin (2011)	
EF551002.1	1	Mitochondrion complete	P. pardus	Not specified	China	Lei et al. (2011)	
EF056501.1	1	NADH 5	P. pardus	Not specified	India	Shouche (2016, unpublished)	
AY035292.1	1	NADH 5	P. pardus	Melas	Java (Indonesia)	Uphyrkina et al. (2001)	
AY035280.1–AY035291.1	12	NADH 5	P. pardus	Shortridgei	Central Africa (Namibia, Botswana, Kruger & Zimbabwe)	Uphyrkina et al. (2001)	
AY035279.1	1	NADH 5	P. pardus	Nimr	South Arabia (wild animal, exact location not specified)	Uphyrkina et al. (2001)	
AY035277.1–AY035278.1	2	NADH 5	P. pardus	Saxicolor	Central Asia (Captive)	Uphyrkina et al. (2001)	
AY035276.1	1	NADH 5	P. pardus	Sindica	Central Asia (Baluchistan)	Uphyrkina et al. (2001)	
AY035270.1–AY035275.1	6	NADH 5	P. pardus	Fusca	India	Uphyrkina et al. (2001)	
AY035267.1–AY035269.1	3	NADH 5	P. pardus	Kotiya	Srilanka	Uphyrkina et al. (2001)	
AY035264.1– AY035266.1	3	NADH 5	P. pardus	Delacouri	East Asia (South China)	Uphyrkina et al. (2001)	
AY035262.1–AY035263.1	2	NADH 5	P. pardus	Japonensis	East Asia (North China)	Uphyrkina et al. (2001)	
AY035260.1–AY035261.1	2	NADH 5	P. pardus	Orientalis	Russia	Uphyrkina et al. (2001)	
KF768352–KF768354	3	NADH 5	P. pardus	Saxicolor	Iran	Farhadinia et al. (2015)	
EF551004.1	1	Mitochondrion complete	Uncia uncia	Out Group	China	Wei, Wu & Jiang (2008)	
EF551003.1	1	Mitochondrion complete	Panthera tigris	Out Group	China	Lei et al. (2011)	
AF385614.1	1	Mitochondrion complete	P. tigris	Out Group	Africa	Dubach et al. (2005)	
HM589215.1	1	Mitochondrion complete	P. tigris	Out Group	China	Zhang et al. (2011)	
AF385613.1	1	Mitochondrion complete	P. leo	Out Group	Brookfield zoo (Exact location unknown)	Dubach et al. (2005)	

Results

We used 43 NADH 5 sequences for the final stage of the analysis. When these were aligned to the 238 leopard sequences available on GenBank, the ML analysis grouped all Pakistani sequences within the Asian leopards, well separated (with a bootstrap of 85% using a K2P model and 52% using a HKY+G model) from both African and Arabian leopards (Fig. 2). While the new sequences obtained in this study could be separated from those of African leopards and attributed to the Asian leopards, the ML analysis showed low genetic variation within the two groups, highlighted by very low bootstrap values throughout the tree (Fig. 2). After confirming that all our sequences belonged to Asian leopards, further analysis aimed to understand the relationships of our samples to the different subspecies of this group. In order to measure genetic variation between our sequences and within the Asian leopards, a haplotype network analysis was performed on a total of 35 sequences (accession numbers MK425702–MK425736). For this analysis, eight samples where discarded due to shorter length of the sequence. Hence, this analysis included 35 samples from this dataset, all the sequences of Asian leopards available on GenBank, a subset of 13 sequences of African leopards and one sequence from an Arabian leopard (Fig. 3). The sequences presented here identified three distinct haplotypes (Figs. 1, 3). The first haplotype (A) included 23 samples and was identical to two sequences previously identified as Panthera pardus fusca (accession number: AY035274.1) and Panthera pardus orientalis (accession number: HQ185550.1) (Fig. 3). The second haplotype (B) included 11 samples with identical sequences to Panthera pardus saxicolor (Fig. 3). The third haplotype (C) was represented by a single Pakistani sample (Fig. 3) and was not the same as any previously identified haplotype. The haplotype diversity amongst the Pakistani samples was based on three segregation sites with a nucleotide variation ranging between 0.003 and 0.006 (Table 3).

Figure 2 Maximum likelihood NADH 5 gene tree (10,000 replicates, K2P model) of 273 sequences from across the globe.

The gene tree separates Asian leopards (green) from Africans (orange), with a bootstrap of 85%. Outgroups are reported in grey.

Figure 3 Median joining network analysis of the Asian leopards.

The network includes 35 sequences of Pakistani leopards, 13 sequences from African leopards, a single sequence from a Persian leopard, and all the sequences from Asian leopards publicly available. Haplotypes are color-coded based on previous subspecies identification. Size of the circles represents the number of sequences with the same haplotype.

Discussion

Despite the limitations of our sample size, this study generated the first genetic datasets for leopard populations in Pakistan. The genetic diversity of leopards from the northern region of Pakistan was determined here based on 43 specimens. As expected, all the samples included in this study clustered within the Asian leopard clade. However, the haplotype network analysis highlighted the presence of three different haplotypes. We recorded a similar nucleotide diversity, ranging between 0.003 and 0.006, to that reported for different leopard subspecies in other studies (Uphyrkina et al., 2001; Farhadinia et al., 2015).

The genetic variation of the Pakistani leopard samples was not consistent with their being a single subspecies in Pakistan, whether existing or novel. Eleven samples (Haplotype B) grouped with sequences of Panthera pardus saxicolor and are here attributed to this subspecies. Similarly, the haplotype represented by a single specimen (Haplotype C) grouped closely to the other P. p. saxicolor haplotypes and is here attributed to this subspecies, as well. The main Pakistani haplotype (Haplotype A), including 23 samples, grouped with two separate sequences previously recorded from two studies in Caucasus (Uphyrkina et al., 2001; Rozhnov, Lukarevsky & Sorokin, 2011). These sequences (HQ185550.1 and AY035274.1) are identical to each other although they were named differently: respectively P. p. orientalis, and P. p. fusca. Based on the similarities between our sequences and the other haplotypes of P. p. fusca, however, we are inclined to consider them as belonging to this subspecies. Similarly, considering the genetic distance between this single sequence of P.p. orientalis obtained from GenBank and the other sequences of the same subspecies, we consider its attribution to orientalis an incorrect identification. However, it is possible that the subspecies P. p. fusca has a large amount of genetic variation, to the point that it could subsume some of the other subspecies. In order to confirm this hypothesis, additional samples from this subspecies are required, together with additional samples of P. p. orientalis and the analysis of additional gene regions.

Table 3 Genetic variation of the Panthera pardus mtDNA from different geographical locations.

Number of samples and number of haplotypes are reported. Haplotype diversity is calculated based on the number of different nucleotides, while nucleotide diversity was obtained from the pairwise distance matrix. The number of segregating sites is reported specifying the number of substitution and if these were transitions or transversions.

Subspecies	Geographic location	Accession number	No of samples	Haplotypes	Haplotype diversity (nt)	Nucleotide diversity	Segregating sites	Transitions	Transversions	Substitutions	
This study	Pakistan	MK425702–MK425744	43	3	1–3	0.003–0.006	3	3	0	3	
P. p. sindica	Pakistan	AY035276.1	1	1	NA	NA	NA	NA	NA	NA	
P. p. saxicolor	Iran	JX559073.1–JX559076.1
HQ185544.1–HQ185548.1
AY035277.1–AY035278.1
KF768352–KF768354	14	4	1–3	0.002–0.004	6	6	0	6	
P. p. fusca	India	AY035270.1–AY035275.1	6	6	2∕3 − 6∕9	0.003–0.01	15	15	0	15	
P. p. kotiya	Srilanka	AY035267.1–AY035269.1	3	2	1	0.002	1	1	0	1	
P. p. orientalis	Russia	HQ185549.1–HQ185550.1
AY035260.1–AY035261.1	4	3	3–6	0.01	7	7	0	7	
P. p. nimer	Arabia	AY035279.1	1	1	NA	NA	NA	NA	NA	NA	
P. p. leo
P. p. shortridgei	Africa	KY292222.1–KY292277.1
JF720187.1–JF720319.1
AY035280.1–AY035291.1	201	30	1 − 20∕23	0.002–0.038	77	75	2	77	
P. p. melas	Indonesia	AY035292.1	1	1	NA	NA	NA	NA	NA	NA	
P. p. delacouri	South Chinese	AY035264.1–AY035266.1	3	3	2–4	0.003–0.01	6	6	0	6	
P. p. japonensis	North Chinese	AY035262.1–AY035263.1	2	2	3/5	0.005	3–5	3–5	0	3–5	

The analysis is consistent with that of Uphyrkina and colleagues who grouped the subspecies P. pardus sindica and P. pardus millardi with the Persian and Indian leopards (Uphyrkina et al., 2001). These two subspecies, first described by Pocock (1930a) and Pocock (1930b) based on morphometric characters only, were retained by Khorozyan and colleagues (2006). On the other hand, all the samples collected for this study likely belong to two subspecies of Panthera pardus: P. p. fusca (Indian) and P. p. saxicolor (Persian), highlighting a higher than expected subspecies diversity for the area that we examined. In fact, our findings are not consistent with the assumption that the river Indus separates the P. p. saxicolor and P. p. fusca subspecies, as it was hypothesised in previous studies (Khorozyan, Baryshnikov & Abramov, 2006; Jacobson et al., 2016). The vast majority of our samples (except those from Baluchistan, Nizampur and Sakher) belonged to the northern region of the Indus, reporting the presence of both P. p. saxicolor and P. p. fusca in this region, where only the latter was hypothesised to live.

The limited number of samples used for this study (especially from the regions of Baluchistan and Nizampur) most certainly does not fully reflect the subpopulation of these regions. Therefore, further studies are likely to discover an even higher diversity in subspecies than that reported here, as already confirmed from Baluchistan, where subpopulations closely related to the Persian leopard are known to be present (Jacobson et al., 2016).

Consequently, our study suggests that the leopards in Pakistan do not appear to have an endemic haplotype or subspecies (Fig. 1). Instead, the Pakistani landscape plays a key role in the ecological overlap of the two leopard subspecies recorded here and it provides suitable conditions for a high level of gene flow between them. Random mating in the expanded region may support the hypothesis that, overall, the Asian population is panmictic and it presents a moderate level of genetic variation (Fig. 2). Panmixis assumes that there are no mating restrictions and, for this to happen, gene flow between leopard sub-populations and sub-species has to happen across all their habitat in Asia. Leopards are known to travel long distances, e.g., 194 km (Fattebert et al., 2013), and have the largest home range recorded for large cats, e.g., 670 km2 (Hunter, 2011) in central Iran.

Pakistan assumes a role in leopard conservation by providing a contact zone for the subspecies P. p. fusca and P. p. saxicolor, subspecies considered endangered (P. p. saxicolor) or near threatened (P. p. fusca). The overlapping distribution of these two subspecies provides an impetus to extend full protection to leopards beyond the limits of regional parks and reserves, over all the Pakistani territory.

The primary constraint in the conservation of leopards in Pakistan is a lack of detailed information, such as presence/absence of each subspecies within the highly fragmented habitat. While the information obtained in this study enables a better understanding of leopard distribution in the areas monitored, further work is required for the more remote regions. Unfortunately, the political instability of some of these regions, together with a widespread lack of financial resources remain a key challenge for sustainable management and conservation of leopards. In addition, illegal hunting of leopards and limited management capacity of local wildlife department in many areas have created an ongoing stressful situation for leopards. This results in leopards moving away from many habitats, such as the Kirthar National Park where they have been reported as locally extinct (Khan et al., 2013).

Conclusions

With the present work, we have highlighted the co-existence of multiple subspecies in the same area in the north of Pakistan. In order to compare these results with other areas of the country, we suggest that further studies focusing on the presence-absence of leopards in Baluchistan, Sindh and in the areas of Punjab and Khyber Pakhtunkhwa are urgently required. In fact, obtaining genetic information from these areas will complement the knowledge now available and enable a better understanding of distribution and ecology of leopards, not only in Pakistan but also worldwide.

Supplemental Information

Supplemental Information 1 The NADH 5 sequences generated for this study

Click here for additional data file.

We thank Anthony Caragiulo and an anonymous reviewer for helpful comments and suggestions on a previous version of this manuscript that helped improving the quality of this work. We are grateful to the Wildlife Department Khyber Pakhtunkhwa Pakistan, Bioresource Research Centre Islamabad, and WWF-Pakistan for providing samples to conduct this research. We thank all the staff at the Wildlife Department Khyber Pakhtunkhwa and WWF-Pakistan for facilitating the study. We also greatly appreciate the assistance provided by Sajid Hussain for collecting and preserving tissue samples from dead leopards killed by communities in retaliation for previous attacks.

Additional Information and Declarations

Competing Interests

Author Contributions

Animal Ethics

Data Availability

The authors declare there are no competing interests.

Muhammad Asad conceived and designed the experiments, performed the experiments, analyzed the data, contributed reagents/materials/analysis tools, prepared figures and/or tables, authored or reviewed drafts of the paper, approved the final draft.

Francesco Martoni analyzed the data, prepared figures and/or tables, authored or reviewed drafts of the paper, approved the final draft.

James G. Ross and Muhammad Waseem approved the final draft.

Fakhar-i- Abbas approved the final draft and provided samples.

Adrian M. Paterson conceived and designed the experiments, analyzed the data, authored or reviewed drafts of the paper, approved the final draft.

The following information was supplied relating to ethical approvals (i.e., approving body and any reference numbers):

Lincoln University Animal ethics committee (AEC) approved this study (AEC2017-02).

The following information was supplied regarding data availability:

Data is available at NCBI GenBank under accession numbers MK425702–MK425744.

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
