# Peer review of "Assessing subspecies status of leopards (Panthera pardus) of northern Pakistan using mitochondrial DNA"

_PeerJ, doi:10.7717/peerj.7243_

## Round 0.1 · original submission · Major Revisions

Thank you for your patience. I'm not sure why, but identifying reviewers who would agree to handle this manuscript was difficult and one of the reviewers that did agree was unable to submit their review until late. Please accept my apologies for the long wait.

Reviewer 1 ·

Basic reporting

A good job overall with clear and professional English for the entire manuscript. Except for a few places, that I had references already in my comments, a good job in general with providing backgrounds and context.

Major comments
1. The introduction can help from background information regarding mtDNA, how it has been used in phylogenetic analyses with proper reference is also going to improve the overall background. Any previous work done with mitochondrial DNA with the cat family/ leopard/ tigers will draw nice relevance to this work as well. (line 83)

2. Why was subunit 5 of NADPH chosen? The authors do not provide enough rationale for this. For the broader audience, it might be beneficial to provide some introductory background explaining the rationale behind the choice? (line 82-83)


3. The format of sequencing, which type, platform etc. is not even mentioned once in the methods. Granted the authors refer the approach to two previous works on lines 114-115, the manuscript might improve if information on the type and nature of sequencing is included here with some details. At the very least- what type of platform was used for sequencing and some statistics on that.


4. The authors mention how this work will impact and contribute “…….to knowledge of species assessment at national and international level and help prioritize the conservation efforts for Pakistan” (Line 88). However, after going through the discussion part, there’s no overall mention of how this work contributes at a global level. The authors do a good job in describing the relevance of the work how it can contribute at the national level mostly in Pakistan, but either remove the international part on the introduction or address that aspect better in the discussion section.


Minor comments

It would be nice to have the Genbank accession number supplemental file referenced somewhere in the manuscript, since they can’t be uploaded to Genbank before July.
No need for switching lines and leaving empty space on line 70

Roberts 1977 missing “et al” ? on line 71-72 ?

Parenthesis needs to be closed for reference; line 80

……Conservation efforts for “leopards in” Pakistan? (line 88)

No need for period/full stops after sub headings (Materials and methods/ introduction etc. as well as after “sample collection”, “DNA extractions and amplifications” etc.)
Any reference for stating leopard population is considered stable in the northern part of Pakistan? (Lines 92-93)

The authors keep switching between numeric and alphabets to describe numbers like 18 and 15. Consistency is needed- choose any one and preferably stick to that pattern. (Lines 95-97)

When mentioning positive results for DNA isolation, mention at least what kind of purity (O.D 260/280 values may be ?) (lines 104-106), the authors obtained and what made the tests positive that convinced the authors to pursue that at a broader scale by outsourcing to South Korea.

Insert comma after Seoul (Line 105)

When stating PCR as abbreviated for the first time (Line 105), consider writing Polymerase Chain reaction here and then abbreviating as PCR in the following lines through the manuscript. The full form instead comes up later on line 113.

The sentence – “These isolated a sequence………….of the mitochondrial DNA.” is confusing and can be written probably in a simpler manner. Please consider revising and rewriting the sentence (Lines 111-112)

Remove period (line 122)
Remove period (line 139)
Remove period (line 160)
Remove colon and may be underline? (line 231) Consistency
Remove period (line 239)
Insert “the” before latter (line 194)

“g” missing from Fig on line 201 and 205

Experimental design

A bigger sample size would have been much better, especially if doing phylogenetic analyses and trying to related to global impact for the study as well.

Validity of the findings

No additional comments. The findings seem to relate to the original hypothesis.

Additional comments

General reporting is good, but authors need to take into consideration some major revisions and some minor revisions to make the manuscript better overall.

·

Basic reporting

1. The title needs to be revised because the research methods do not address subspecies status in Pakistan. See Point 1 from "Experimental Design" comments.
2.The Results and Discussion sections are disjointed and would benefit from more continuity between the sentences. This would make it easier to understand for an international audience and follow the logic of the authors.
3. The manuscript would benefit from a broader context for the work. The authors do a good job detailing the scarcity of genetic data on leopards in Pakistan, but need more info to frame why finding subspecies within Pakistan is important. It would also be good to talk about any geographic barriers that would either limit or enhance movement of leopards into and within Pakistan, as this would directly influence isolation and the evolution of potential subspecies/populations.

Experimental design

1. The research methods don't address the statement of the title. The title suggests the manuscript surveyed the majority of Pakistan and looked at multiple mitochondrial DNA markers to assess genetic diversity of leopards across all of Pakistan. The manuscript had a very limited geographic scope and only looked at one mitochondrial DNA locus to determine if haplotypes within the NADH-5 gene matched leopards from other geographic origins.
2. The manuscript needs to be reframed to address the limited geographic scope of sampling (northern Pakistan) and the use of a single mitochondrial DNA locus. Such instances occur in lines 33-34 (authors present no evidence Pakistani leopards are experiencing a bottleneck), lines 161-162 (not a genomic dataset because only uses mtDNA, and can't say Pakistani populations because only looked at northern Pakistan), lines 180 - 183 (authors may have excluded genetic information from gene regions other than NADH-5, and looking at these areas may uncover more information).

Validity of the findings

1. The authors should explain why samples were too poor in quality for the haplotype analysis, but not too poor in quality for the phylogenetic analysis (line 149). Missing or incomplete data could have effected the ML trees.
2. The use of a single mitochondrial DNA locus and limited geographic scope limit what the authors can say and they overstate much of their findings as already outlined in this review. The results are still compelling, but the scope of the manuscript's findings must be more properly stated to reflect these shortcomings.

Additional comments

1. The methods described are sound and the results are valuable, however the main issue is the statements regarding the implications are overblown. Sequencing multiple nuclear markers (such as microsatellites or nuclear genes) in addition to more mitochdronail markers would have been better suited to answering questions of gene flow and bottlenecks. Without this information, the authors are limited to describing how northern Pakistani leopards compare to other leopard samples with respect to the NADH-5 gene. The paper should be reframed as such.

---

## Round 0.2 · accepted · Accept

Thank you for addressing the reviewer comments so thoroughly.